# An Evidence-Based Update on the Molecular Mechanisms Underlying Periodontal Diseases

**DOI:** 10.3390/ijms21113829

**Published:** 2020-05-28

**Authors:** Syed Saad B. Qasim, Dalal Al-Otaibi, Reham Al-Jasser, Sarhang S. Gul, Muhammad Sohail Zafar

**Affiliations:** 1Department of Bioclincial Sciences, Faculty of Dentistry, Kuwait University, Safat 24923, Kuwait; 2Department of Periodontics and Community Dentistry, College of Dentistry, King Saud University, Riyadh 60169, Saudi Arabia; dalalotaibi@ksu.edu.sa (D.A.-O.); raljasser@ksu.edu.sa (R.A.-J.); 3College of Dentistry, Periodontics Department, University of Sulaimani, Sulaymaniyah 1124-30, Iraq; sarhang.hama@univsul.edu.iq; 4Department of Restorative Dentistry, College of Dentistry, Taibah University, Al-Madina Al-Munawwarah 41311, Saudi Arabia; mzafar@taibahu.edu.sa; 5Department of Dental Materials, Islamic International Dental College, Riphah International University, Islamabad 44000, Pakistan

**Keywords:** periodontitis, molecular mechanism, inflammation, genetics, proteomics, metabolomics, saliva

## Abstract

Several investigators have reported about the intricate molecular mechanism underlying periodontal diseases (PD). Nevertheless, the role of specific genes, cells, or cellular mechanisms involved in the pathogenesis of periodontitis are still unclear. Although periodontitis is one of the most prevalent oral diseases globally, there are no pre-diagnostic markers or therapeutic targets available for such inflammatory lesions. A pivotal role is played by pro- and anti-inflammatory markers in modulating pathophysiological and physiological processes in repairing damaged tissues. In addition, effects on osteoimmunology is ever evolving due to the ongoing research in understanding the molecular mechanism lying beneath periodontal diseases. The aim of the current review is to deliver an evidence-based update on the molecular mechanism of periodontitis with a particular focus on recent developments. Reports regarding the molecular mechanism of these diseases have revealed unforeseen results indicative of the fact that significant advances have been made to the periodontal medicine over the past decade. There is integrated hypothesis-driven research going on. Although a wide picture of association of periodontal diseases with immune response has been further clarified with present ongoing research, small parts of the puzzle remain a mystery and require further investigations.

## 1. Introduction

Periodontitis is the sixth most prevalent human disease globally and has been a challenging disease for clinicians around the globe. With a high prevalence (11.2%), periodontitis has been estimated to affect more than 743 million individuals [1,2]. The understanding of the etiology and pathogenesis of periodontal diseases (PD) has evolved for decades. Therefore, this dynamic inflammatory condition is not fully understood in terms of underlying mechanisms at the molecular level. There are several questions that still need to be addressed: What triggers the shift from the localized and contained inflammatory response of gingivitis to progressive and destructive periodontitis? At what instance and how does the subgingival microbiome become dysbiotic? What is the temporal relationship between the dysbiotic microbiome and the innate and acquired immune response? Is it the bacterial invasion of tissues that is an initiator or consequence of the disease [3]? The current model of pathogenesis is outlined by the critical interplay among biofilm bacteria (colonizing micro-organisms), host’s genetic factors (immune inflammatory), and stressors acquired environmentally [4]. In addition to the bacterial challenges [5,6], dysregulated synthesis of inflammatory mediators are present behind the major pathogenicity and tissue destruction [7]. These factors lead to periodontitis that is characterized by irreversible histopathological alterations that clinically appear as bone loss, loss of tooth attachment apparatus due to the destruction of periodontal ligaments (PDL), and ultimately tooth loss [8]. According to Van Dyke et al., a factor that has eluded researchers for decades is the identification of a true infectious pathogen for periodontitis [3]. The associations of periodontitis with various systemic diseases including cardiovascular diseases [9,10], cancer [11], rheumatoid arthritis [12], and diabetes [13] are well established.

Periodontal medicine saw significant advances during the revolutionary era of molecular medicine of the 1950s [14]. This revolution displaced the classical physiological and whole organism biology. It led to a progressive era towards reductionist science, which was able to deliver amazing insights into the complex chronic diseases such as oral host microbiome disequilibria and associated pathological conditions. Hypothesis-driven research on periodontal pathogenesis has identified the importance of oral microbiome dysbiosis and host immunity to periodontal pathogens [14]. According to Sima and Van Dyke, while this reductionist research has revealed the complexity of intracellular signaling pathways, it has somehow managed to miss the larger picture by not combining the major areas of biological hierarchy, such as gene, protein, and lipid networks; metabolomics; and intracellular signaling pathways [14].

Nevertheless, the role of specific genes, cells, or cellular mechanisms underlying the pathogenesis of periodontitis is unclear. Therefore, until today, there are no pre-diagnostic markers or therapeutic targets for this disease. Previous studies [15,16,17] have reported inconsistent findings with respect to genetic predisposition to periodontitis [18]. An enhanced conceptual model of the pathogenesis of periodontitis was put forward by Kornman with respect to omics technologies, namely genomic, proteomic, or metabolomic correlations, to harness the dynamic nature of the biochemical processes in periodontitis [19]. The aim of the current review is to deliver a piece evidence-based update on the molecular mechanism of periodontitis with a particular focus on recent developments. The current review also delivers an extensive evidence-based update about the role of inflammatory markers including associated omics technologies (genomic, transcriptomics, proteomics, and metabolomics) for diagnosing, treatment, or prognosis. Furthermore, recommendations are put forward for future research areas that can aid to understand the pathophysiology of periodontitis at a molecular level.

## 2. Evolving Concepts in Host Modulation

The host inflammatory response is what largely drives the pathological process [6]. It involves a physiological response to bacterial infections and the initiation of healing processes. As periodontal inflammation progresses, a variety of pro-inflammatory cells such as eosinophils, neutrophils, and macrophages (Figure 1) are released. This intricate biological process in turn is balanced with anti-inflammatory agents at the same time. When tissues are damaged by microbial invasion, it triggers antigen presentation in the form of antigen-presenting cells (APCs) such as dendritic cells (DCs); thereafter, the inflammatory immune reactions is triggered [20]. On the other hand, to effectively enhance anti-bacterial responses, DC receptors are sometimes used as targets for such purposes. For instance, myeloid DCs are used as targets for enhancing anti-bacterial immune responses to capture antigens and activation of T-cells, which modulates other immune cells by changing the behavior of both parenchymal and endothelial cells as well as any periodontal pathogens associated with lipopolysaccharide (LPS) [21]. LPS are also present in the cell wall of the Gram-negative bacteria and they are responsible for triggering host response. Recognition of innate immune signals by DC relies on a limited number of pathogen related receptors. Toll-like receptors (TLRs) are among the proteins responsible for regulating apoptosis, inflammation, and immunity. Clinically healthy gingiva also expresses certain TLRs that are responsible for playing certain role in the periodontal inflammatory response [22].

Studies [21,22,23,24] have also elucidated that activated DCs are often matured enough to amplify the initial activating signal provided to T cells from the T cell receptor and pave way for the development of distinct cytokine patterns. For instance, the production of characteristic cytokines (Figure 1) such as INF-*γ* (interferon-gamma) and IL (interleukin)-17 is very common. Cytokine patterns are often essential in setting up selective migration patterns and life cycles of CD4 T cells [25]. However, since CD4 T cells play crucial roles in mediating a variety of cytokine’s production patterns, they may differentiate into one of several lineages of T helper cells. Zhu et al. (2009) [26] mentioned that, during TCR activation, CD4 T cells might differentiate into Th1, Th2, Th17, and Treg based on the production patterns of cytokine. Each CD4 T cells has a role to play in the healing processes. For instance, Th1 cytokines, particularly IFN-γ and IL-2, are responsible for controlling immune responses occurring within the cells, including cell apoptosis, which is associated with further periodontal tissue damage. The Th2 cytokines (Figure 1), IL-4 and IL-10 in particular, enhance the anti-inflammatory factors as well as the overall immunity of an individual [25].

Moreover, many activities take place to protect the host from pathogenic elements and foreign organisms that can cause infection. Immune surveillance is a general term that is used in describing how the molecular patterns are formed to prevent pathogen-related infections. Thus, all these activities happen when inflammation becomes chronic enough to damage the tissue. A compelling piece of evidence also indicates that the Th17 lineage cells play a role in protecting mucosa against pathogens, and enhancing anti-inflammatory responses [27]. However, with respect to pathogenesis in chronic periodontitis, the roles of DCs and T-helper cells have not been clarified efficiently in all aspects. Both DC and T-helper cells might initiate a periodontal inflammatory cascade since they possess the characteristics of biologic mediators as well as the reactive oxygen species (ROS) which trigger the inflammatory cycle [28]. Therefore, when they are excessively activated, they may result in the development of the periodontal inflammatory cascade. Hence, their roles remain controversial. Another example is of the role of nitric oxide (NO), which is one of the host inflammatory mediators that is highly toxic when used excessively as host defense [25].

### 2.1. Cytokines, Proteases and Prostaglandins

Healthy periodontium maintains a complex balance between pro- and anti-inflammatory cytokines, and disruption of this balance in favor of pro-inflammatory cytokines results in periodontal tissue damage [22]. These cytokines (Figure 1) are often produced through the stimulation of the host cells in contact with the dental biofilm. Tumor necrosis factors (TNF) and IL are two prominent cytokine families that are secreted during this process; however, other products that originate from the acute phase such as arachidonic metabolic acid (AA) and other complementary elements such as thromboxane and prostacyclin are excreted as well. Smith et al. (1993) provided an example by monitoring Rhesus monkeys with the use of ligature-induced periodontitis model for over six months. They reported substantial changes in gingival crevicular fluid (GCF) levels of PGE_2_, TxB_2_, IL-1*β*, TNF-*α*, and Leukotriene B4 (LTB4) [29]. It was also proven that, during the inflammatory processes, several molecules stimulated bone resorption such as IL-6, macrophage colony-stimulating factor (MCSF), and prostaglandins-2 (PG-2) [30,31]. IL-6 and IL-1*β* were also found to be the most potent cytokines that stimulate bone resorption through activating RANK ligand and hence promote osteoclast activity (Figure 2) [32,33,34].

On the other hand, it was also revealed that the gene family with IL-1 cytokine had three different members with different receptor antagonist: IL-1*β,* IL-1*α,* and IL-1 [36]. The catabolic events of these cytokines are managed through endogenous inhibitors that embody IL-1 and TNF receptor antagonists. When administered for healing capabilities, those antagonists can decrease infection [37]. Using the receptors from a particular cytokine tends to limit pathogens from developing further disease progression. This means that the processes of developing effective interventions on therapeutic modalities will require thorough investigations to gain insights on the available inflammatory mediators as well as other factors that need to be elucidated. In conclusion, using cytokine antagonists to treat PD in this scope needs to be further examined properly to obtain conclusive and effective treatment modalities. Recently, Rath-Deschner et al. reported higher levels of CXCL_1_ (motif chemokine ligand 1), CCL_2_ (motif chemokine ligand 2), and CCL_5_ (motif chemokine ligand 5) chemokines in vitro and in vivo in human and rat gingiva in sites with periodontitis [38]. CXCL_1_, also known as GRO-alpha, is a potent chemoattractant for neutrophils [39], whereas CCL_2_ and CCL_5,_ also known as MCP-1 and RANTES, are primarily responsible for the recruitment of monocytes, macrophages and T-lymphocytes [38].

### 2.2. Pathways of Tissue Destruction via MMPs

In zinc-dependent proteolytic enzymes and their related families, matrix metalloproteinases (MMPs) play a crucial role in regulating pathophysiological and physiological processes in repairing damaged tissues, reshaping of cytokines, remodeling extracellular matrix, and activating defensins and mediators [40] MMPs also mediate and regulate immunomodulatory responses by releasing cytokines that prioritize growth factors in favor of cells and tissues, as well as modify nonmatrix substrates such as chemokine to enhance the healing processes [41].

MMPs can also be indicated in any issue that is related to the destructive tissues and/or pathological conditions, as well as periodontitis defensive processes, wound healing, etc. This is specifically related to MMP-2, -9, and -8 [41]. Besides, MMPs share a common gene with at least 24 families of distinct gene characteristics and similar structural features that are related to Zinc enzymes possessing [42]. Based on the structural and functional features, MMPs can be categorized into collagenases, stromelysin, gelatinases, matrilysins, and membrane-type (MT) collagenases [42]. Since MMPs are active members of the proteinase family, they usually become active on the surface of the cell or in the intracellular spaces after being synthesized. However, on the cell surface, MT-MMPs often demonstrate their activities, which differ from other soluble MMPs. This means that they share different cellular origins and substrate specificity despite having a common gene [43] It has also been argued that the collagenase species of MMP-8 and/or inductive collagenase-2 are the major ones found in active sites of periodontitis rather than MMP-1, which is demonstrated constitutively as a typical example of a fibroblast-type [44]. This is because fibroblast-type MMP-1 usually plays the role of remodeling tissues since it is linked to the processes of restructuring tissues to their normal states. Besides, there is another subfamily member of MMPs called MMP-13 or collagenase-3, which tends to affect the epithelial and fibroblasts part of gingiva [45]. Moreover, collagenase-3 (MMP-13) has been identified in GCF of subjects with periodontitis [46].

### 2.3. RANK/RANKL/OPG Interactions

An important mechanism influencing the development of periodontal disease involves bone resorptions, which is influenced by the RANK, RANKL, and OPG system (Figure 2) [23,47]. RANK, the activation receptor of NF-kB, was detected on osteoclasts. Its ligand, RANKL, is a transmembrane protein expressed in different cell types, especially osteoblasts and activated T cells. RANKL is known to force the cell to interact physically with the precursors of osteoclasts and cementoclasts and binds to its receptor (RANK) to induce bone resorption during the course of periodontitis. On the other hand, Osteoprotegerin (OPG), a member of the TNF family [48], is expressed by osteoblasts, cementoblasts, fibroblasts, and T lymphocytes [49]. This acts as an antidote to inhibit bone resorption through binding to its ligand (RANKL), hence preventing the latter from binding to its receptor (RANK) [50].

Therefore, when this can be presented in a clinical scenario, the progression of periodontal disease was associated with increased levels of cells that are positive markers of bone resorption (RANK and RANKL) and a decrease of the RANKL ligand (OPG) [51]. Recent evidence indicates that the production of RANKL occurs by activation of T lymphocytes present in the tissue in states of inflammation, modulating osteoclastogenesis and bone resorption that lead to bone loss in periodontitis. RANKL expression was higher in advanced periodontal disease when compared with gingivitis [52]. This suggests that RANKL plays an important role in destruction of periodontal tissues and its inhibition might decrease the resorption of periodontal bone [53]. This fact can be reinforced by the decrease in OPG (RANKL ligand and osteoclastic modulator) observed in periodontitis [54].

Overall, RANKL/OPG ratio can be considered the main determinant of bone metabolism during the progression or regression of PD. Bone resorption is stimulated by both the increased RANKL level and the decreased OPG level which acts as pro-resorption signals (Figure 2) [55]. Currently, RANKL inhibitors or OPG boosters as therapeutic agents for the treatment of PD are promising, but are still under investigation [46,56].

### 2.4. Evidence-Based Updates on RANK, RANKL and OPG

Several studies in humans and mice support the concept that Th1 cells and their cytokines characterize early/stable periodontal lesions [26,57,58]. On the other hand, Th2 cells are associated with disease progression, consistent with the B-cell nature of the progressive lesion. Other studies, however, have attributed destructive effects to Th1 cells (Figure 2). Recent evidence indicates that PD are not adequately describable in simple Th1 vs. Th2 dichotomous terms. In this regard, the discovery of the Th17 subset may lead to a more nuanced understanding of host–pathogen interactions in the periodontium [25,34]. The implication of Th1 cells in destructive inflammation in various diseases should be re-interpreted, since the presence or involvement of IL-12, a signature Th1 cytokine, was assessed by evaluation of the IL-12/23 common p40 subunit, and hence the involvement of the Th17 subset cannot be ruled out [59]. Current approaches in clinical periodontal practice focus primarily on decreasing bacterial challenge rather than modulating the host response. Under the extended Th1/Th2/Th17 paradigm, it may be feasible to elucidate what constitutes protective vs. destructive host response in periodontitis [25]. It would then be possible to develop therapeutic intervention modalities to maximize the protective and minimize the destructive aspects of the periodontal T-cell responses. Other mediators being investigated for modulation include nuclear factor kappa B and endothelial cell adhesion molecules [39,60]. However, the role of these inflammatory mediators in periodontitis needs to be elucidated.

## 3. Genomic and Transcriptomic Association

Genomic technologies, including transcriptomics, proteomics, and metabolomics, have provided new prospective to understand the pathogenesis of PD and host–microbial interaction [61]. Transcriptomic data show significant differences in expression of several transcripts among healthy, chronic, and aggressive periodontitis subjects. The genes associated with immune response and natural killer cell receptors (IL-6 and selectin E) were upregulated [4]. Interestingly, a significant increase in gene expression of viruses has been reported in periodontitis subjects compared to healthy subjects. Furthermore, genes expression related to lytic phage were upregulated in the saliva of subjects with PD [62]. This highlights the possible role of viruses in PD.

Proteomic studies have been conducted to find a proteomic signature of healthy [63] and diseased periodontium [64] in GCF, saliva, and gingival tissues. Histones are new markers identified using proteomic analysis, and it is a nuclear DNA binding protein that important in the organization of DNA structure and inflammatory response. Histones are found in large amounts in subjects with gingivitis and periodontitis, whereas barely detected in healthy subjects [65,66,67]. Moreover, actins and keratins are found in periodontitis, explaining the rapid epithelial turnover that happened by bacterial invasion, Actin, on the other hand, is associated with osteoclastic activity, indicating further bone loss [41,68]. Other proteins, such as cystatins (protease inhibitors) [66] and fibronectin [69], are detected at higher levels in periodontitis patients.

Metabolomic analysis has also used to identify metabolites as possible new markers and new metabolic mechanisms related to PD. Aggressive and chronic periodontitis contributed to the high amount of tyrosine, phenylalanine, and proline, and low amount of lactate, pyruvate, and N-acetyl groups. Gingival crevicular fluid in aggressive periodontitis subjects showed an elevation in noradrenaline, ribose, dehydroascorbic acid, lysine, and xanthine levels and reduction in glutathione, 2-ketobutyric acid, glycine-d5, and thymidine levels. This suggests metabolic markers of oxidative stress, purine degradation, tyrosine, and pyrimidine metabolism are elevated. The levels of antioxidant vitamins (vitamins A, C, and E) and redox-active metals (calcium, magnesium, zinc, copper, iron, and selenium) were significantly decreased in periodontitis patients and their levels inversely proportioned with the severity of PD [4]. Cadaverine and hydrocinnamate are considered as highly specific biomarkers of PD and their salivary levels were significantly reduced after periodontal treatment [4].

The development and functioning of various human body tissues are controlled by genetics. A gene stores information for the synthesis of a specific protein that, in turn, controls multiple body mechanisms and functions. Therefore, a defective gene may lead to an inevitable aberration in the physiological functioning and even the development of pathologies [70]. The genomic approaches have been applied for the prevention or curing of several conditions including cancers, infectious diseases, and genetic and autoimmune disorders [70]. Gene therapy has also emerged as an attractive concept for restoring the oral tissues affected due to trauma, dental caries, and PD by applying basic approaches including in vivo and ex vivo gene therapy [71]. For example, the ex vivo gene therapy based on multipotent dental stem cells with the potential to differentiate into any dental tissues. Successful regeneration of the periodontium has been accomplished using a combination of stem cell engineered by adenovirus to express the BMP-2 gene [72].

The association of genetics with a number of PD is well evident. Schaefer et al. [73] reported the association of aggressive periodontitis with rs1537415, present in the glycosyltransferase gene (*GLT6D1*). In another recent study, Munz et al. [74] reported the association of *SIGLEC5* (rs12461706) with aggressive periodontitis. Mutations in the NOD2 gene identified by whole-exome sequencing resulted in aggressive periodontitis [75]. Similarly, a number of studies [76,77] reported genetic involvement in chromic periodontitis among various populations. A rare *TSNAX*-*DISC1* RNA polymorphism (rs149133391) showed a correlation with chronic periodontitis among Hispanic/Latino residents [77]. Hong et al. [76] reported the association of *TENM2* (*ODZ2*) as with chronic periodontitis among Koreans. Some components, such as defensin alpha-1 and alpha 3 polymorphisms (rs2978951, rs2738058) have been associates with aggressive periodontitis [74] as well as chronic periodontitis study [78]. Recently, Hiyari et al. conducted a genome-wide association study using an animal model and identified CXCR3 as a locus of bone loss and associated lipopolysaccharide induced periodontitis [79].

With the development of genetic and sequencing technology, transcriptome data may reveal altered gene expression associated with PD [4]. Several studies [63,80] have investigated the gene expression profiles associated with periodontitis and identified the defensive downregulation/upregulation genes linked to bone resorption and inflammatory response. Demmer et al. [80] used gene ontology to compare the gingival tissues biopsy and reported a remarkable upregulation of most of the chemokines, TNF signaling, platelet-derived growth factor (PDGF), cell adhesion molecules, c-Jun N-terminal kinase (JNK), p38, and mitogen-activated protein kinase (MAPK) pathways in the inflamed gingival tissues compared to the control. At the same time, there was a downregulation of various components of the transforming growth factor-beta (TGF-β) family in the inflamed tissues [80]. These findings were further verified by the RNA sequencing of periodontitis tissues confirming the upregulation of pathways associated with immunity proteins, B-cell receptor signaling, and cytokine and chemokine activities in periodontitis tissues [63]. In addition, periodontitis resulted in a significant downregulation of desmocollin1 gene that is obligatory for the epithelial cells adhesion and desmosome formation [80]. Simultaneously, gingivitis tissues showed a suppressed level of various keratin genes (keratin1, keratin2, and keratin3) that ultimately may influence the epithelial barrier by inhibition of cytokeratin proteins in epithelial spinous layer [64]. Therefore, in the case of periodontitis, the reduction in the genes regulating epithelial homeostasis may compromise the mucosal epithelial barrier, facilitating the infiltration of plaque pathogens/metabolites and inducing an inflammatory response in the connective tissues and alveolar bone resorption [4]. In addition to the diminished epithelial barrier defense, another contributory transcriptomic factor is interleukin-17 (IL-17) pathway upregulation in periodontitis [63,80]. IL-17 belongs to a cytokine family that is secreted by immune cells, including lymphocytes and plays a defensive role against pathogens [81,82]. IL-17 shows complex interactions with human tissues; the upregulation of IL-17 may lead to pathological mucocutaneous diseases such as lupus erythematosus, psoriasis, and ulcerative colitis [83,84]. On the contrary, a lack of IL-17 activity compromises the defense mechanism; the disruption of IL-17 signaling due to mutation makes individuals vulnerable to fungal infections [85]. Accordingly, balanced homeostasis of IL-17 is required to maintain the mucosal defense and periodontal health [4]. Although the involvement of epigenetic and genetic elements in the development of periodontitis is well evident, the current research and understanding of the associated mechanisms are still at an infancy level. The real challenges are to explore various epigenetic pathways regulating complex diseases and associated therapeutic genetic approaches. A better understanding of factors such as recognition of related genes, vectors, and delivery at the site is required for precise cellular therapy [71].

### Proteomics and Peptidomics Approaches to the Disease

Peptides and proteins are biomolecules that are essentially present in the living organisms and required for proper various body functions [86,87]. There is a variety of proteins present in the human body that vary structurally from each other according to their specific functions [87,88]. The science of proteomic deals with the comprehensive analysis of proteins or proteomes to understand their characteristics for physiological and pathological states [89]. For the analysis of PD at the proteomic level, GCF and peri-implant fluid are available (Figure 3). Both fluids contain extensive in-build information on a systemic level. The collection of these biofluids is challenging and needs special care to avoid contamination from microbes for better analysis. Currently, paper-based sample collection method is available called Perio paper. After collection, the sample fluid is used in a range of analytical techniques including mass spectrometry [90,91], protein purification [92,93], polyacrylamide gel electrophoresis [90,94,95], matrix-assisted laser desorption ionization [96,97], nuclear magnetic resonance [98,99], and high-performance liquid chromatography [100,101] for proteomic analysis. The combination of proteomic analysis remarkably helped in understanding the pathological basis of various diseases, including periodontitis [89]. Many researchers [102,103,104] have reported the proteomic analysis of various PD including gingivitis [103], chronic periodontitis [105], and aggressive periodontitis [102,104,106] by analyzing saliva [103,105], gingival crevicular fluid (GCF) [106,107], and gingival tissues [108,109]. Although proteins are abundant in saliva and GCF, the later contains biomarkers associated with PD and provides site-specific analysis [89]. However, the proteomic analysis of GCF is challenging due to certain limitations of GCF such as limited volume [110] and the presence of albumin [111,112]. In addition, the concentration of a number of biomolecules (such as cytokines) is very low in the GCF [113,114] and may not be detected using mass spectrometry [115]. Monari et al. [109] investigated gingival tissues for proteomic analysis and reported the presence of an abundance of proteins associated with periodontitis.

In the case of periodontitis, GCF may identify various human proteomes, including immunoglobulins, apolipoproteins, cytoskeletal proteins, and neutrophil-derived extracellular histones [108,118,119]. The proteomic analysis identified biomarkers specific chronic periodontitis included nuclear DNA-binding proteins (histones), antimicrobial proteins, and extracellular matrix proteins [110]. Histones (H2A, H2B, and H4) (Figure 3) are usually absent in periodontally healthy subjects but appear in abundance in chronic periodontitis patients [113,114,115]. Besides, Grant et al. [120] analyzed GCF and identified cytoskeletal-related proteins (actins and keratins) corresponding to cell destruction by bacterial invasion in periodontitis. The presence of actins is indicative of osteoclastic activity and bone resorption [120]. The proteome analysis may have variations depending on the progress of periodontitis; protease inhibitors (such as cystatins) are identified during the initial stages of periodontitis [113], while the secretion of glycoproteins (fibronectin) enhances during the progression of periodontitis [121]. The presence of proteins biomarkers related to neutrophil-secreted proteins, cellular proliferation, migration, and adhesion are associated with aggressive periodontitis [104]. Recently, Hartenbach et al. [116] reported data about the salivary proteomic analysis of chronic periodontitis patients (Figure 3a,b). In periodontitis patients, the salivary acidic proline rich phosphoproteins, a submandibular salivary gland androgen regulated protein, and a cystatin were increased (Figure 3b). Among others, histatin-1, fatty acid binding protein, and thioredoxin were predominant in the diseased state [116]. Shin et al. conducted deep sequencing using proteomic and detected S100A8 and S100A9 as potential biomarkers for periodontitis [117] (Figure 3c).

The periodontitis management by standard mechanical debridement alters the secretions proteins and inflammatory mediators [105]. It is worth emphasizing that the main focus of proteomics in periodontics is to analyze various biomarkers associated with PD in terms of diagnosis and prognosis.

## 4. Investigations and New Molecules Discovered in Immunochemical Pathways of Periodontitis

New studies revealed that PD and periodontitis, in particular, are caused mainly by dysbiosis of bacterial community, rather than an infection caused by an individual or a group of bacteria [62,122]. This new concept pushed researchers to explore the mechanisms of how this dysbiotic microbiota interacts with the host that alternatively leads to the initiation and progress of the PD. This part highlights the most up to date molecules and pathways that have shown to be contributed to PD [3,4].

The complement system mediates immune surveillance as well as hemostasis of tissue in collaboration with physiological and other immune systems [82,123]. When dysregulated, however, by microbial virulence and host genetic factors, the complement will turn to pathogenic factor that drives several inflammatory diseases including periodontitis [124,125]. Dysregulation of the complement system has been shown to affect dental biofilm on the one hand and the immune system on the other hand. On dental biofilm, using C5a receptors-1 by *P. gingivalis* results in a numerical and compositional change of microbiota (dysbiotic community) that, in turn, causes periodontitis in susceptible subjects [126,127]. Activation of C5a receptor-1 and toll-like receptor (TLR)-2 interference in macrophages and neutrophils has a deleterious effect on the cellular immune response. In neutrophil, this crosstalk inhibits antimicrobial response by ubiquitination and proteasomal breakdown of the TLR-2 adaptor by releasing transforming growth factor-beta-1 that enhances differentiation of myeloid primary response protein-88 ubiquitination. Phagocytosis by neutrophil is also inhibited by activation of phosphoinositide 3-kinase and suppression of actin polymerization and RhoA GTPase [31]. In macrophage, NO synthase-dependent pathogen killing is suppressed via activation of C5a receptors-1. The activation of C5a receptors-1 leads to weak cyclic adenosine monophosphate (cAMP) responses as the activation happens only by TLR-2, thereby suppressing glycogen synthase kinase-3 beta and nuclear factor-kappa B; thus, it suppresses the killing of the pathogen by inducing NO synthase-dependent pathway [31]. Overall, the complement system and C5a receptor in particular play an important role in PD progression via dysbiosis and inflammation, mainly by an effect on neutrophil [128].

Developmental endothelial locus-1 (Del-1) (Figure 4) is another molecule that is related to PD. It affects the transmigration of circulating neutrophils to peripheral tissue in response to infection. Neutrophil’s extravasation requires firm adherence of neutrophils to endothelium [129,130]. This critical adhesion occurs between inter-cellular adhesion molecule-1 on endothelial cells and activated lymphocyte function-associated antigen-1 on neutrophils. Del-1 controls this adherence by binding to the latter molecule, thus inhibiting excessive neutrophil transmigration to infection and inflammatory sites in normal condition [131,132]. This explains the strong association of leukocyte adhesion deficiency type 1 due to lack of Del-1 and aggressive form of periodontitis [59,133] and the use of Del-1 in the treatment of excessive infiltration of neutrophils in diseases such as periodontitis [131,132].

Sirtuins is another family of biomolecules that contributes to PD [137]. Sirtuins involve seven proteins (sirtuin 1–7) that are located in the cytoplasm, mitochondria, or nucleus. Nuclear sirtuins 1, 2, 6, and 7 have deacylation activity and are mainly associated with several inflammatory diseases, including periodontitis. For example, sirtuin 1 can inhibit B cells activity via nuclear factor kappa light chain, thereby reducing inflammatory mediators expression of cytokines (such as TNF-alpha, IL-1β, -6, and -8), NO synthase and cyclooxygenase-2 [138]. Activated sirtuins 1 is related to inflammation and apoptosis, and it can be suppressed by Zn^++^ binding agents such as hydroxamic acid and MMP inhibitor [139,140]. This suggests the potential use of non-antibiotic tetracyclines and herbal products such as curcuminoids as host modulation therapy.

Inflammasomes are multimeric protein structures composed of a sensor molecule and have shown to be involved in PD recently. Inflammasomes are usually activated by neutrophils, monocytes, macrophage, and Dcs [139]. Activation of inflammasomes is highly inflammatory, and this activation is usually firmly controlled to stop abnormal activation [24]. Inflammasomes are essential in the maturation of pro-inflammatory cytokines such as IL-1β and IL-18 that play a significant role in inflammation through the recruitment of immune cells, mainly neutrophils, B cell activation, and antibody production, as well as T cell differentiation [141]. The role of inflammasomes in periodontal health still needs to be further investigated. However, data reveal higher expression of inflammasome components in experimentally induced periodontitis as well as in GCF, saliva, and gingival tissue of individuals with periodontitis [141,142]. Inhibition of inflammasomes via suppressing upregulation of intracellular signaling pathways, inflammasome components blockage, and inhibiting inflammasome-mediated cytokines (Il-18 and IL-1β) is useful in PD treatment [141].

Resolution of inflammation is another new biochemical process which is important in acute inflammation cessation and preventing the progression to chronic inflammation, fibrosis, and tissue scarring; thus, it enhance regeneration of diseased tissues [143,144]. Lipid mediators such as lipoxins, resolvins, protectins, and maresins, which are considered as special pro-resolving mediators, play roles in the inflammatory resolution, not only by controlling white blood cells function, but also exhibiting receptors that control osteoclast and stem cells that differentiate to fibroblast and osteoblast, which are important in preventing tissue damage during inflammation [145,146] and improve tissue regeneration and bone healing in periodontitis [147,148]

Finally, periodontitis is reported to be related with some novel genes. Desmocollin 1 gene that encodes calcium-dependent glycoprotein that is mainly found in epithelium is essential in desmosome formation and adhesion of cells. In periodontitis, there is significant downregulation of this gene compared to healthy control [80]. Furthermore, cytokeratin proteins found in the upper spinous layer of epithelial cells are important in the formation of epithelial cell barriers. These cytokeratin proteins are encoded by keratin family genes (keratins 1–3), which are significantly restrained in the gingival tissue of diseased individuals [64]. Suppression of these genes in periodontitis subjects suggests compromised mucosal and gingival barrier that alternately lead to easy penetration of pathogen and their metabolic products to the underlining connective tissue. Genes associated with neural expression recognized as another common activation pathway during inflammation, including periodontitis. During inflammation, neural chemokine genes, prokineticin-2, and Kallmann syndrome-1 are highly expressed and have a wide range of communications with other pathways of epithelial barriers, immune response, wound healing, and vascular system [33,149]. The role of the neural process in PD is not well appreciated and further studies are required to elaborate this part. In summary, the involvement of various molecules in periodontal disease initiation and progression reflects the complex nature of this dysbiotic disease as a result of host–bacterial interaction. Studies examining the role of pathogens, immune response, and genomic technologies are molding our understanding of disease processes.

## 5. Clinical Implication

In contemporary dental clinics, the diagnosis of periodontal diseases solely relies upon clinical measures (probing depth, clinical attachment loss, and bone loss). However, these measures only provide limited insights about past periodontal disease activity rather than the current state of the disease and future disease progression or likely outcome of periodontal treatment [110]. Furthermore, in an epidemiological survey where many subjects need to be examined, taking full mouth measures of these variables is very challenging. Thus, an attempt has been made to enhance diagnostic and prognostic capabilities through using the molecules that contribute to the pathogenesis of periodontal diseases. Among the mentioned molecules, MMP8 has been translated as a point of care chairside test [150,151]. MMP8 has been shown to be associated with the severity of the diseases and can predict the treatment outcome [152]. MMP8 chairside test can be used to determine subjects that at high risk for further disease progression and subjects that will not respond to non-surgical periodontal treatment [153]. In clinical practice, this will be helpful in reducing patient overtreatment, identifying those patients at high risk for further disease progression, and improving treatment outcome by providing complex periodontal therapy only to those patients with high levels of MMP8 with chairside test. On the population level, as MMP8 is associated with the severity of periodontal diseases [152], the MMP8 chairside test can be very useful to identify early detection of periodontitis in epidemiological investigations. This could also be applied to several biomarkers including Interleukin 1 (IL-1), Interleukin-6 (IL-6), and others. These biomarkers can be measured from soft tissue biopsies taken from associated deep periodontal pockets, from saliva samples, or GCF [154,155]. On the other hand, knowing the molecular mechanism underlining periodontal diseases leads to the introduction of low dose doxycycline (Periostat^®^ and Oracea^®^) as host modulation therapy for aiding periodontal therapeutics. The host modulation therapy decreases the expression of inflammatory cytokines; inhibits MMPs; and enhances collagen synthesis, osteoclastic activity, and bone formation [139].

Further studies are necessary to translate clinical significance of these findings into practice, such as the role of genetic background in subject susceptibility of the disease, identifying the high-risk group through the molecular mechanism underlining the diseases, and preventing the disease progression by interfering in between the molecular cascade.

## 6. Conclusions and Future Perspectives

These new emerging molecules and their association with PD processes require further studies to integrate and harmonize all these data to better understand the initiation and progression of PD and determine risk assessment for each individual. Recent reports regarding the molecular mechanism of PD have revealed unforeseen results indicative of the fact that significant advances have been made to the periodontal medicine over the past decade. There is integrating hypothesis-driven research going on along with systems medicine investigations of chronic PD. Promises made by omics to advance precision medicine and personalized care are at the initial stages to validate the road map to elaborate integration into clinical practice. However, there are still certain aspects that need addressing in term of large-scale applications of such intricate techniques to understand periodontal osteoimmunology. Additionally, it is essential to elaborate that, although a wide picture of association of PD with immune response has been further clarified with present ongoing research, small parts of the puzzle remain a mystery and require further investigations.

## Figures and Tables

**Figure 1 ijms-21-03829-f001:**
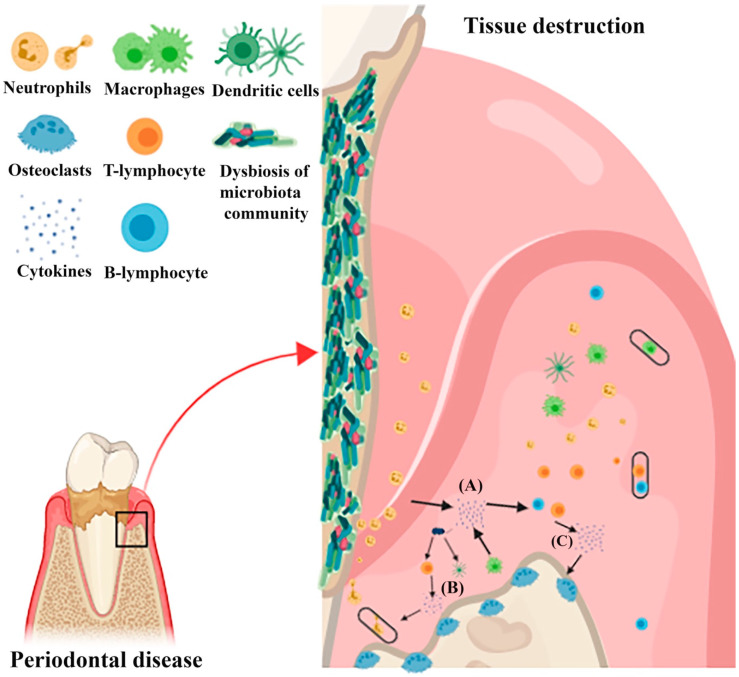
Diagrammatic illustration depicting pathogenicity of periodontitis and the interplay of cytokines involved. Pathogenicity of locally present microbiota is enhanced by the colonization of pathogens, thereby over activating the immune response and triggering tissue destructions. (**A**) The first wave of cytokine secretions that takes part in amplifying the pro-inflammatory cytokine cascade and aid in recruitment, activation, and differentiation of specific immune cells is shown. (**B**) Additionally, a group of cytokines closely correlated to differentiation of a specific subset of lymphocytes are secreted by mononuclear phagocytes and antigen presenting cells after stimulation by microbiome. (**C**) Each of these cells releases a certain pattern of cytokines which might act as the positive feedback loop, finally leading to destruction of the tissue.

**Figure 2 ijms-21-03829-f002:**
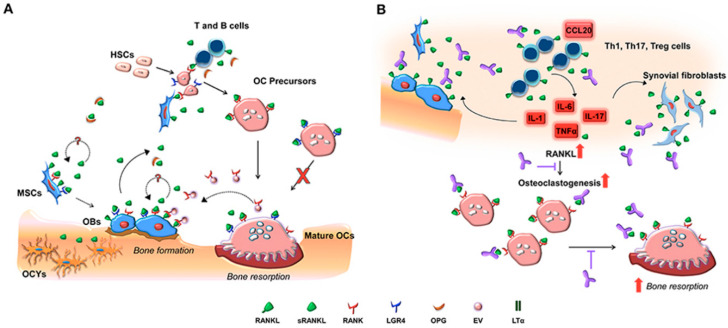
Diagrammatic illustration of molecular and cellular players taking part in RANK, RANKL, and OPG signaling in bone during physiological (**A**,**B**) pathological conditions. Soluble RANKL synthesized by osteoblasts and immune cells induce osteoclastogenensis when attaching to RANK on osteoclast precursors [35]. OPG is the soluble decoy receptor for RANKL. The expression of RANK by mesenchymal stem cells and osteoblasts points to a potential RANKL autoregulatory mechanism affecting bone deposition. Furthermore, extracellular vesicles (EV) initiate a reverse signaling on osteoblast (**B**), depicting enhanced synthesis of RANKL by immune cells and osteoblastic cells [35]. This exaggerates osteoclast generation and bone loss. Image adapted with kind permission from the publisher.

**Figure 3 ijms-21-03829-f003:**
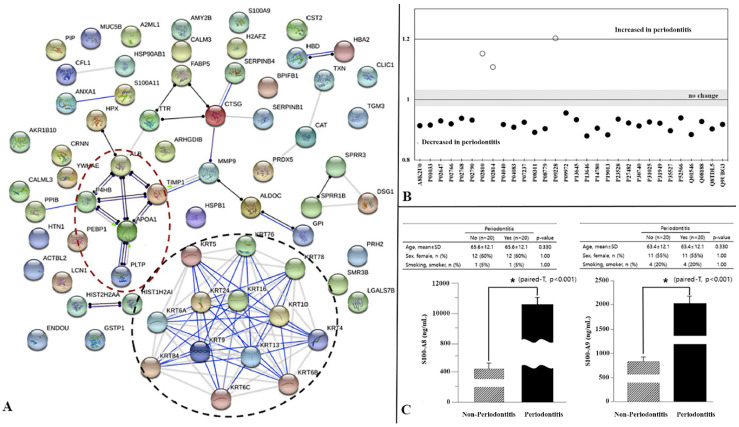
(**A**) Hartenbach et al. reported the protein–protein interaction network analysis of all salivary proteins involved in health and diseases state of periodontium. Black lines depict that one protein reacts non-specifically with another. Blue lines indicate there is binding among them. The purple lines mean catalysis. (**B**) The salivary protein difference between periodontal health and chronic periodontitis [116] (**C**) Shin et al. studied S100A8 and S100A9 using ELISA as potential candidate biomarkers [117]. The images are adapted with permission from the publisher.

**Figure 4 ijms-21-03829-f004:**
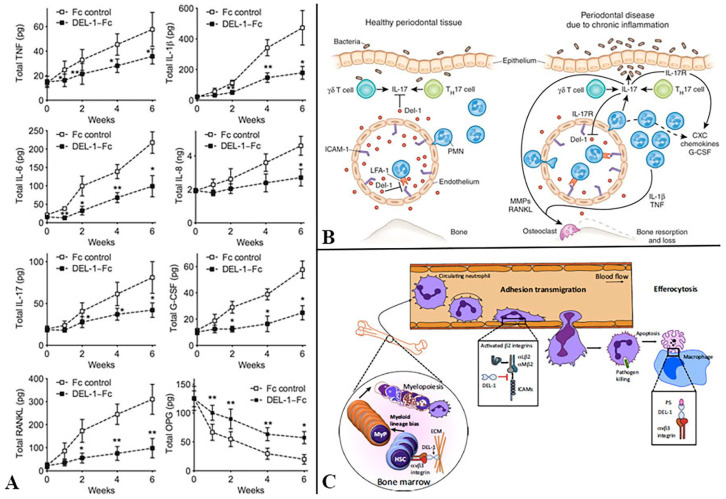
Reports about Del-1: (**A**) Shin et al. reported inferior amounts of pro-inflammatory cytokines of Del-1–Fc-treated monkeys. Cytokine analysis of GCF showed that Del-1–Fc treatment resulted in significantly lower levels of proinflammatory cytokines, whereas OPG, was sustained at higher levels in Del-1–Fc-treated sites [134]. (**B**) Diagrammatic illustration by Khader showing healthy periodontal tissue and diseases state. A higher expression of Del-1 in young mice hinders IL-17 synthesis and neutrophil trafficking, whereas, in older mice, Del-1 expression is downregulated and IL-17 production by γδ T cells, CD4^+^ T cells, and neutrophils is upregulated. IL-17 induces G-CSF and CXC chemokines to attract neutrophils into the inflamed tissue [135] (**C**) Hajishengallis and Chavakis reported that Del-1 interacts with β3 integrin on hematopoietic stem cells (HSCs) and induces HSC proliferation and biased differentiation toward the myeloid lineage (MyP, myeloid progenitors). In the vessel lumen, Del-1 blocks the interaction between LFA-1 (aLb2) integrin on neutrophils and ICAM-1 on the vascular endothelium, thereby inhibiting neutrophil adhesion and transmigration. Ddl-1 can also block Mac-1 (aMb2) integrin, which mediates intraluminal crawling of neutrophils, although whether Del-1 inhibits crawling has not been specifically addressed [136]. All images adapted with kind permission from the publisher.

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
