# Peer review of "An Evidence-Based Update on the Molecular Mechanisms Underlying Periodontal Diseases"

_ijms, 2020, doi:10.3390/ijms21113829_

Round 1

Reviewer 1 Report

The review presented gives a good, deep and novel overview and summary of the current evidence on the topic. So, I would like to congratulate the authors in their effort put in preparing the manuscript.

Author Response

Dear Reviewer, 

The authors of the manuscript would like to thank the reviewer for such encouraging comments 

Regards

Authors

Reviewer 2 Report

This review, by Qasim SB et al., tried to provide evidence-based update on the molecular mechanisms underlying periodontal diseases.  The authors have tried to be thorough and inclusive for compiling most of the PD related research results and publications.  Although it is quite convincing by reading the review that significant advances have been made to the periodontal medicine over the past decade, the clinical significance of these findings has been missing from this write up.  With all these know factors contributing to the understanding of PD, is any of the markers/factors used in diagnosis and/or preventing PD in a clinical setting?  If yes, what are they?  If not, how far away? What’s the foreseeable blocks that are preventing this from research bench to clinical translation?  The review is very long but without those answers it’s not very insightful.

I also have some minor critics:

  • Line 40, molecular mechanism should not be separated with a “;”
  • Line 52, “is the bacteria……initiator”, fix the English
  • Line 55, “uncontrolled synthesis”, should be “dysregulated synthesis”
  • Line 89, “molecules such as ……”, the examples given are different types of leukocytes, therefore, these are cells, not molecules
  • Line 91, “when tissues are damaged……” here should be microbial invasion so to initiate antigen presentation
  • Line 136, “this is because;……” fix English sentence
  • Line 147, spell out “IL”
  • Line 160 to 163, where is this section come from? Which reference?

Author Response

Re: Response to Reviewer comments for the review titled “An evidence based update on the molecular mechanism underlying periodontal disease,"

Sincerest thanks for your response and reviewer comments on our manuscript. The authors have modified the manuscript according to the reviewer's comments. All text changes have been represented as track changes and easy to follow. Please find below a point-by-point response to the reviewer’s concerns. We hope that you find our responses satisfactory and that the manuscript is now acceptable for publication.

Reviewer #2: The review by Qasim SB et al., tried to provide evidence based update on the molecular mechanisms underlying periodontal diseases. The authors have tried to be thorough and inclusive for compiling most of the PD related results and publications. Although it is quite convincing by reading the review that significant advances have been made to the periodontal medicine over the past decade, the clinical significance of these findings has been missing from the write up.

Comment 1. With all these know factors contributing to the understanding of PD, is any of the markers factors used in diagnosis and or preventing PD in clinical setting. If yes, what are they? In not how far away? What’s the foreseeable block that is preventing this from the research to bench to clinical translation? The review is very long but without those answers, it’s not very insightful

Response: We thank the reviewer for the comment. The authors agree with the reviewer and therefore a section about clinical implications has been added to the existing text. It now reads as follows

“ 5. Clinical implication

In contemporary dental clinics, the diagnosis of periodontal diseases solely relies upon clinical measures (probing depth, clinical attachment loss and bone loss). However, these measures only provide limited insights about past periodontal disease activity rather than the current state of the disease and future disease progression or likely outcome of periodontal treatment [110]. Furthermore, in an epidemiological survey where a large number of subjects need to be examined, taking full mouth measures of these variables is very challenging. Thus, attempt has been made to enhance diagnostic and prognostic capabilities through using the molecules that contribute to the pathogenesis of periodontal diseases. Amongst mentioned molecules, MMP8 has been translated as a point of care Chairside test [150, 151]. MMP8 has shown to be associated with the severity of the diseases and can predict the treatment outcome [152]. MMP8 chair-side test can be used to determine subjects that at high risk for further disease progression and subjects that will not respond to non-surgical periodontal treatment [153]. In clinical practice, this will be helpful in reducing patient overtreatment, identifying those patients at high risk for further disease progression and improve treatment outcome by providing complex periodontal therapy only to those patients with high levels of MMP8 with chair-side test. On the population level, as MMP8 associated with the severity of periodontal diseases [152], the MMP8 chair-side test can be very useful to identify early detection of periodontitis in epidemiological investigations. This also can be implied on several biomarkers including Interleukin 1 (IL-1), Interleukin-6 (IL_6), and others. These biomarkers can be measured either in from soft tissue biopsies taken from associated deep periodontal pockets, from saliva samples or GCF[154, 155]. On the other hand, knowing the molecular mechanism underlining periodontal diseases leads to the introduction of low dose doxycycline (Periostat®, Oracea®) as host modulation therapy for aiding periodontal therapeutics. The host modulation therapy decreases the expression of inflammatory cytokines, inhibits MMPs, enhances collagen synthesis, osteoclastic activity, and bone formation [156].

Further studies are necessary to translate the clinical significance of these findings into practice, such as the role of genetic background in subject susceptibility of the disease, identifying the high-risk group through the molecular mechanism underlining the diseases, and preventing the disease progression by interfering in between the molecular cascade.”

  1. Sorsa T, Alassiri S, Grigoriadis A, Räisänen IT, Pärnänen P, Nwhator SO, et al. Active MMP-8 (AMMP-8) as a grading and staging biomarker in the periodontitis classification. Diagnostics. 2020;10:61. doi:10.3390/diagnostics10020061.
  2. Räisänen IT, Heikkinen AM, Pakbaznejad Esmaeili E, Tervahartiala T, Pajukanta R, Silbereisen A, et al. A point-of-care test of active matrix metalloproteinase-8 predicts triggering receptor expressed on myeloid cells-1 (TREM-1) levels in saliva. J Periodontol. 2020;91:102–9. doi:10.1002/JPER.19-0132.
  3. Gul SS, Douglas CWI, Griffiths GS, Rawlinson A. A pilot study of active enzyme levels in gingival crevicular fluid of patients with chronic periodontal disease. J Clin Periodontol. 2016;43:629–36.
  4. Gul SS, Griffiths GS, Stafford GP, Al-Zubidi MI, Rawlinson A, Douglas CWI. Investigation of a Novel Predictive Biomarker Profile for the Outcome of Periodontal Treatment. J Periodontol. 2017;88:1135–44.
  5. Huang W, He BY, Shao J, Jia XW, Yuan Y Di. Interleukin-1β rs1143627 polymorphism with susceptibility to periodontal disease. Oncotarget. 2017;8:31406–14.
  6. Liu Y-CG, Lerner UH, Teng Y-TA. Cytokine responses against periodontal infection: protective and destructive roles. Periodontol 2000. 2010;52:163–206. doi:10.1111/j.1600-0757.2009.00321.x.
  7. Golub LM, Lee HM. Periodontal therapeutics: Current host-modulation agents and future directions. Periodontol 2000. 2020;82:186–204.

Comment 2. Line 40 molecular mechanism should not be separated with a “;”

Response, Thank you for the comment, “:” has now been deleted in between molecular mechanism (Line 40).

Comment 3. Line 52, “is the bacteria …… initiator” fix the English

Response; Thank you for the comment. We have modified the sentence and it now reads as

“….or is it the bacterial invasion of tissues that is an initiator or consequence……” (Line 52)

Comment 4. Line 55. “ uncontrolled synthesis “ should be “dysregulated synthesis”

Response; The term “ uncontrolled synthesis “ is now changed to “ dysregulated synthesis” (Line 56).

Comment 5. Line 89 “ molecules such as….” The examples given are different types of leukocytes therefore these are cells not molecules.

Response: The new statement reads as “ pro-inflammatory cells such…” (Line 90).

Comment 6. Line 91 “ When tissues are damaged… “ here should be microbial invasion so to initiate antigen presentation

Response: The new statement now reads as “When tissues are damaged by microbial invasion, this event triggers antigen presentation in the form of antigen-presenting……” (Line 92 to 93).

Comment 7. Line 136 “ this is because…..” fix English

Response: The new sentence now reads as “. Both DC and T-helper cells might initiate a periodontal inflammatory cascade since they possess…” (Line 142).

Comment 8. Line 147, spell out “IL”

Response: Interleukin has been spelled out at the first instance it appears (Line 125).

Comment 9. Line 160 to 163 where is this section from which references?

Response: We thank the reviewer for pointing this out. A reference has been added to missing and  subsequent section

“……. Leukotriene B4 (LTB4) [29]. It was also proven that during the inflammatory processes, several molecules stimulated bone resorption such as IL‐6 , macrophage colony‐stimulating factor (MCSF) and prostaglandins-2 (PG-2) [30, 31]. IL‐6 and IL‐1β were also found to be the most potent cytokines that stimulate bone resorption through activating RANK(Figure 2) ligand and hence promote osteoclast activity [32–34].

On the other hand, it was also revealed that the gene family with IL-1 cytokine had three different members with different receptor antagonist. They include; IL‐1β, IL‐1α, and IL‐1 [35]. The catabolic events of these cytokines are managed through a manner of endogenous inhibitors that embody IL-1 and TNF receptor antagonists. When administered for healing capabilities, those antagonists can decrease infection [36].”

The authors have responded to all comments raised by the reviewer and revised the manuscript according to the best of their knowledge. We hope that the revised manuscript will satisfy the reviewer and will be suitable for publication in IJMS. once again thank the Editor and all the reviewers to have contributed their precious time in reviewing this. Thank you very much. 

Yours sincerely,

Dr.Syed Saad B Qasim

Round 2

Reviewer 2 Report

The revised version is considerably improved.  I have no further comments.